energy/power and energy systems/engineering geology

dam stability, variable weight theory, TOPSIS model, comprehensive evaluation model, bow tie model, safety assessment

**Author for correspondence:**
Fuliang Jiang
e-mail: jfljfd@163.com

# Comprehensive evaluation system for stability of multiple dams in a uranium tailings reservoir: based on the TOPSIS model and bow tie model

Fuliang Jiang[1,2,3], Haonan Wu[1], Yong Liu[1,2,3], Guan Chen[1], Jintao Guo[1] and Zhe Wang[1]

[1]School of Resource & Environment and Safety Engineering, University of South China, Hengyang, Hunan 421001, People's Republic of China
[2]Hunan Province Engineering Technology Research Center of Uranium Tailings Treatment Technology, Hengyang, Hunan 421001, People's Republic of China
[3]Hunan Province Engineering Research Center of Radioactive Control Technology in Uranium Mining and Metallurgy, Hengyang, Hunan 421001, People's Republic of China

 FJ, 0000-0002-2391-9795; HW, 0000-0003-1892-8715

The main purposes of this study are to analyse the evaluation of tailings dam stability under multiple factors and prevent accidents more effectively by proposing a composite risk analysis model. The evaluation model combining the TOPSIS model and bow tie model is presented in this paper. Firstly, a new formula was adopted to calculate the integrated weights based on the subjective and objective weights and the theory of the TOPSIS model was introduced. Secondly, taking a uranium tailings reservoir in south China as an example, the index values and constant weights of the 10 dams are determined according to eight aspects of accumulating dam crest elevation, dam slope, mechanical properties, seepage capacity, topographical conditions, flood control capacity, rainstorm resistance capacity and earthquake resistance capacity. Thirdly, the fitting degree between the stability and ideal solution of each dam is calculated by using the TOPSIS model. The stability fitting degree of the 10 dams is 76%, 93%, 82%, 90%, 66%, 79%, 85%, 96%, 32%, 89% in sequence. This result shows that among the 10 dams, the 9[#] dam ranks the lowest in stability. The actual results are in good consistency with those calculated by the TOPSIS model, which can provide a scientific and reliable new idea for the safety of other multi-index comprehensive evaluations. It is worth mentioning that it can still maintain high accuracy of dam stability evaluation under multiple indexes and multiple dams. Also, the comprehensive evaluation model proposed

in this paper can more effectively reflect the subtle differences between similar evaluation objects. Fourthly, safety barriers from both the cause of the accident and the consequences was established by using the bow tie model to block the path of the accident, and to propose safety measures to make the evaluation system more complete.

## 1. Introduction

In recent years, the number of tailings ponds in China has increased rapidly with the booming mining industry. Meanwhile, accidents caused by unstable tailings ponds are also becoming more serious. For example, on 25 January 2019, the dam of tailings pond which belongs to Companhia Vale do Rio Doce in Brazil broke, resulting in 179 deaths and 131 missing; on 8 September 2008, a serious dam-break accident occurred in the tailings reservoir of Xiangfen County, Shanxi Province, resulting in 277 deaths and 96.192 million yuan of direct economic losses. According to statistics, there were more than 80 tailings pond accidents from 2007 to 2017 [1]. A uranium tailings pond is a special kind of tailings pond, which is the largest radioactive waste storage site in the nuclear fuel cycle system and a potential long-term radioactive pollution source. Once a dam break occurs, the consequences will be more serious. At present, there are approximately 1 billion cubic metres of uranium tailings produced in the world, and the number of uranium tailings reservoirs is 186. There are many tailings reservoirs storing radioactive materials in China. The uranium content in uranium tailings is from 4 to 10 times higher than in the natural background in the general soil, and life of a nuclide in more than 1000 years accounts for about 30% [2,3]. The impact of radioactive sources on the ecological environment, especially water and soil, is very important [4]. With the increase of uranium tailings ponds, the corresponding environmental pollution problems are becoming more and more serious [5], and the attention is getting higher and higher socially. It can be seen that ensuring safety and reliability of uranium tailings ponds is of great significance to the safety of people's lives and property and the sustainable development of the ecological environment.

The stability of tailings ponds has always been a key indicator for assessing its safety, as is the case with uranium tailings ponds. At present, the main methods for evaluating the stability of tailings ponds are: the cloud model analysis method [6,7], the set pair analysis method [8,9], the fuzzy comprehensive evaluation method [10,11], the extension evaluation method [12,13] and so on, but there are few studies on stability evaluation of multiple dams, and the accuracy of the results is relatively low. When comparing the stability of different dams in the same tailings pond, it is necessary to involve multiple factors such as economic input, site selection, natural conditions and mine dam itself. It is necessary to comprehensively consider various indicators and reasonable analysis indicators. The influence weight of stability can make the result closer to the actual situation of the project. Although the system engineering method can solve the one-sided decision caused by considering only a single factor, there is still no reasonable basis for determining the weight value of the influencing factors [14]. When solving problems with constant weights, the weight vector is fixed regardless of the state of each factor. This often leads to unreasonable comprehensive results in practical problems, that is, the 'state imbalance' problem. The idea of variable weight is to adjust the weight vector dynamically, not only considering the order of relative importance of various basic factors, but also considering the degree of state equilibrium, so as to solve the problem of 'state imbalance' to some extent [15]. Hegde & Das combined qseudo-static seismic loading with the strength reduction analysis to check the seismic stability of the dam, and conducted nonlinear dynamic stability analysis to simulate the true earthquake events [16]. Tatu & Stematiu evaluated the safety of the tailings dam in Novat, and the unexpected rise of the water table in the pond was the main cause of the dam failure. The results show that in both hypotheses, the hydraulic gradient at the dam crest is higher than the critical hydraulic one (0.54), which leads to the internal erosion of the dam body failure [17]. In order to find a rapid evaluation method for the stability of rock slope, Jiang et al. constructed a TOPSIS evaluation model based on entropy weight and made some improvements to meet the evaluation needs. Finally, four sections of slope were selected as examples to study, and the results were compared with those of improved grey relational evaluations and extension evaluations, and the three evaluation results are consistent [18]. Dong et al. used a strength reduction method to analyse the difference of slope stability under different rainfall and rainfall duration. The results show that there was a positive correlation between rainfall and the overall displacement of the slope, and the stability evaluation of the slope under rainfall conditions should take factors such as slope deformation characteristics, seepage law and safety factor into comprehensive consideration [19].

In this paper, a multi-index evaluation model is established by combining variable weight theory with the TOPSIS model [20,21]. By means of a variable weight index and decision model, the stability of the uranium tailings dam is sorted to find the dam with weak stability in the uranium tailings pond. The accuracy of the method is verified by comparing the evaluation results with those obtained by the traditional Swedish circle method. Then, after briefly analysing the weak links using activity-based classification (ABC) analysis, the bow tie model is used to analyse the causes and propose safety countermeasures. The evaluation results can not only provide reference for preventing the instability of tailings dams and guide the decommissioning of uranium tailings reservoirs, but also provide a reference and a new method for safety evaluation of the uranium tailings dam group.

# 2. Methods

## 2.1. Evaluation index system construction

The stability of a tailings pond is influenced and determined by many complex factors such as its internal structural parameters, external environment and safety facilities. Among them, both external load and internal weak links have great uncertainty [22]. Therefore, in order to establish a scientific and reasonable evaluation index system, it is necessary to fully understand the evaluation objects, consult relevant literature and find out the representative key indices of each part to represent the overall stability of the tailings pond.

## 2.2. Determination of weight based on variable weight theory

### 2.2.1. Determining variable weight vector

In the view of the problem that the constant weight cannot reflect the actual situation accurately, the paper adopts the variable weight theory to determine the weight. According to the relevant definitions of variable weight theory [23–25], for any constant weight vector $W = (w_1, \ldots, w_n)$, the variable weight vector is obtained through equalization based on the following formula:

$$W(X) = \frac{W \cdot S(X)}{\sum_{j=1}^{n} [w_j \cdot S_J(X)]}. \tag{2.1}$$

In the above formula, $S(X)$ is the equilibrium state function, when $x_i \geq x_j \to S_i(X) \geq S_j(X)$, $S(X)$ is the excitation state function, which means the weight is positively correlated with the indicator state; when $x_i \geq x_j \to S_i(X) \leq S_j(X)$, $S(X)$ is the suppression state function, which means the weight is negatively correlated with the indicator state.

$W \cdot S(X) = [w_1 S_1(X), \ldots, w_n S_n(X)]$ is named the Hardarmard product.

### 2.2.2. Constructing the equalization function

According to the literature, the state variable weight vector is the gradient vector of some $m$-dimensional real function, which is called the equilibrium function [26,27]. If the increase of the value produces a favourable positive effect on the results, then the variable weight vector is of an excitation type; if the increase of the value has adverse effects on the results, then the variable weight vector is of the penalty type. According to the two types of variable weight vector, there are two kinds of equilibrium functions: penalty type and incentive type.

The specific steps of constructing the equalization function are as follows: (i) selecting the appropriate function type, and considering its application conditions, advantages and disadvantages; (ii) determining the state value of each factor, and considering its influence on the weight; and (iii) selecting the appropriate coefficient, and considering the indicator balance requirements.

This paper refers to the relevant literature of variable weight theory to select the exponential state vector and achieves based on the following formula:

$$S(X) = \begin{cases} e^{-\alpha(x_{ij} - \beta)}, & x_{ij} \leq \beta \\ 1, & x_{ij} \leq \beta, \end{cases} \tag{2.2}$$

where $\alpha$ represents the penalty level ($\alpha \geq 0$); and $\beta$ represents the negative level ($0 < \beta \leq 1$). Here $\alpha$ is taken as 0.5, $\beta$ is taken as 0.1, and calculated by combining the constant weight vector.

## 2.3. Construction of the TOPSIS model

The basic principle of the approximate ideal solution ranking method (TOPSIS) is to sort the evaluation objects by the distance between the positive ideal solution and the negative ideal solution in multi-objective decision-making problem to get the best scheme [28].

### 2.3.1. Establishing the multi-index evaluation matrix

The evaluation matrix is determined by two elements including the evaluation object and the evaluation index. According to the two, a multi-index evaluation matrix is constructed:

$$A = \begin{bmatrix} a_{11} & a_{12} & \cdots & a_{1j} \\ a_{21} & a_{22} & \cdots & a_{2j} \\ \vdots & \vdots & \ddots & \vdots \\ a_{i1} & a_{i2} & \cdots & a_{ij} \end{bmatrix}.$$

### 2.3.2. Evaluation matrix normalization

Because the selection of evaluation indicators needs to be as comprehensive and representative as possible, each indicator comes from different subsystems, and there is no unified unit and dimension, so it is impossible to make direct comparison. Therefore, it needs to be normalized for subsequent calculation. The processing methods are as follows:

$$X_{ij} = \frac{a_{ij}}{\sum_{i=1}^{m} a_{ij}}. \tag{2.3}$$

### 2.3.3. Variable weight normalization matrix construction

The variable weight matrix $W$ and the normalized evaluation matrix $X$ represent the index weight and evaluation index, respectively. By multiplying the corresponding terms of the variable weight matrix $W$ and the normalized evaluation matrix $X$, the weighted evaluation matrix is obtained as follows:

$$C = \begin{bmatrix} w_{11}x_{11} & w_{12}x_{12} & \cdots & w_{1n}x_{1n} \\ w_{21}x_{11} & w_{22}x_{22} & \cdots & w_{2n}x_{2n} \\ \vdots & \vdots & \ddots & \vdots \\ w_{m1}x_{m1} & w_{m1}x_{m1} & \cdots & w_{mn}x_{mn} \end{bmatrix}.$$

### 2.3.4. Evaluation object fitting degree calculation

The ideal solution can be determined as follows:

$$C^+ = \left\{ (\max_i C_{ij} | j \in J^+), (\min_i C_{ij} | j \in J^-) \right\} \tag{2.4}$$

and

$$C^- = \left\{ (\min_i C_{ij} | j \in J^+), (\max_i C_{ij} | j \in J^-) \right\}, \tag{2.5}$$

where $C^+$ and $C^-$ represent positive and negative ideal solutions, respectively; $J^+$ represents the set of dominant indicators, the larger the index value is, the closer it is to the ideal; and $J^-$ represents the disadvantage indicator set, and the smaller the indicator, the better.

Calculating the ideal solution distance:

$$d_i^+ = \sqrt{\sum_{j=1}^{n} (C_{ij} - C_j^+)^2} \tag{2.6}$$

and

$$d_i^- = \sqrt{\sum_{j=1}^{n} (C_{ij} - C_j^-)^2}, \tag{2.7}$$

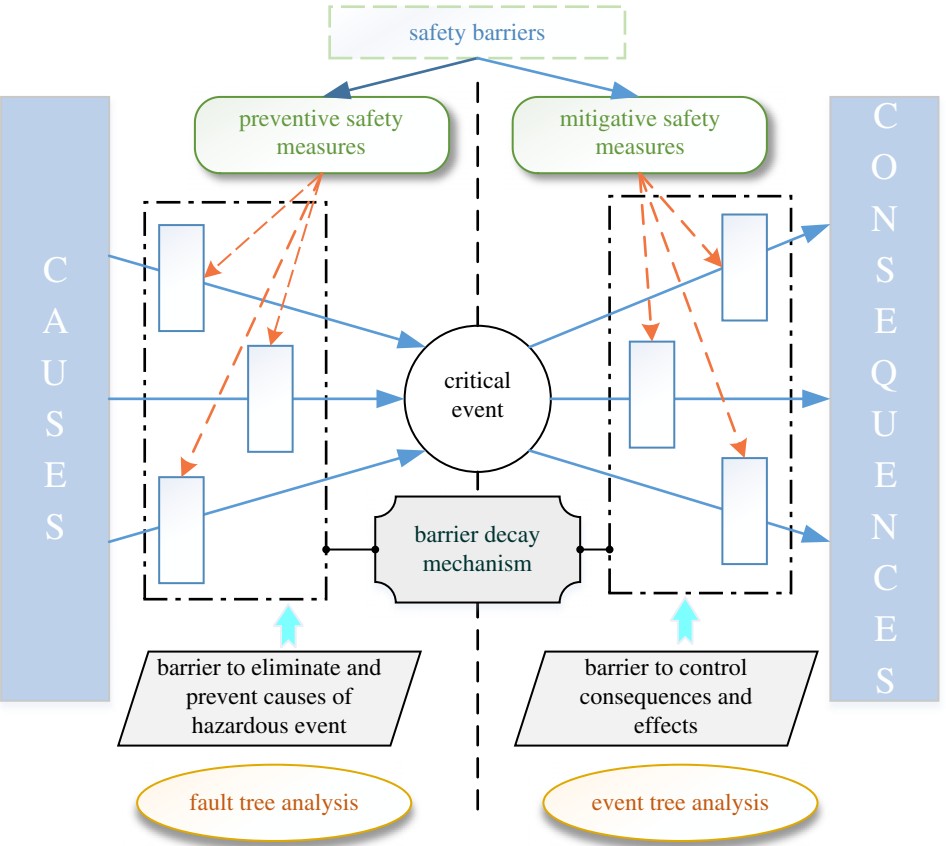

**Figure 1.** Bow tie model sketch.

where $d_i^+$ and $d_i^-$ represent the distance between the evaluation object and the ideal solution, respectively; and $C_i^+$ and $C_i^-$ represent the corresponding positive and negative ideal solution values, respectively.

The fitting degree between the evaluation object and the ideal solution can be calculated as follows:

$$E_i^+ = \frac{d_i^-}{d_i^- + d_i^+}, \quad 0 \leq E_i^+ \leq 1, \tag{2.8}$$

where $E_i^+$ represents the fitting degree between the evaluation object and the positive ideal solution. The greater the value of $E_i^+$, the greater the advantage of the evaluation object.

## 2.4. Activity-based classification (ABC analysis method)

The ABC analysis method is a basic work and cognition method in the field of economy. By identifying a few key factors that play a decisive role in things and most of secondary factors that have little impact on things, it can reasonably allocate time and power to save costs and improve economic benefits [29]. A class A factor represents the main influencing factor; a class B factor represents the secondary influencing factor; and a class C factor represents the general influencing factor. Because of its simple and easy to use characteristics, this method was originally used in the field of inventory control. Later, some scholars combined it with various methods to continuously improve [30–32], which made ABC analysis widely used.

## 2.5. Bow tie model

As a safety assessment method, the bow tie model (figure 1) is an effective way to propose pertinent measures for the critical event set in the central box. The causes of the critical event are indicated on the left of the bow tie and consequences on the right. The causes are the events which may lead to accident and its principle is similar to fault tree analysis, while the consequences are losses of life and property by the accident [33,34]. According to the causes and consequences of the critical event, safety barriers need be set up to block the path of the critical event and prevent it from happening. In

order to prevent accidents effectively, preventive safety measures are set on the left side of the bow tie to prevent the critical event; mitigative safety measures are placed on the right side to minimize the consequences of the critical event; safety barrier functions are to prevent, control, or mitigate undesired events or accidents. If a barrier function is performed successfully, it should have a direct and significant effect on the occurrence or consequences of an undesired accident [35]; a barrier system is a barrier group in which multiple safety barriers collectively implement or perform a barrier function, and it is inappropriate to include all barriers in a single level. Besides the title that is most often used 'escalation factor' contributes to many and poorly understood errors. Therefore, the preferred title is 'barrier attenuation mechanism', because it is more explicit about the hierarchy of barriers rather than all of them being included in the main path [36].

# 3. Results

## 3.1. Application

A uranium tailings pond in south China is located in a densely populated area. The tailing sand and waste water of the tailing pond contain more than 10 kinds of radioactive materials such as radium, uranium, thorium and other heavy metal pollutants. There is also a large area of soft soil substratum in the uranium tailing pond, and the average thickness of titanium white mud is 4–20 m. In total, a huge area needs to be treated. The tailings pond is located in the upper reaches of the Hengyang section of Xiangjiang River. The largest residential area nearby are the "272" and "710" factories. In addition, the stability of the tailings ponds is consequent upon the nearby residents' safety and the Xiangjiang Rivers' sustainability.

The circumference of the tailings pond in the south is 6000 m, which is surrounded by 10 dam sections, whose length is 4600 m, accounting for about 77% of the total length of the pond circumference. The total slope area of the dam section is about 220 000 m$^2$. The names of the dam sections are: the first column dam (1$^\#$ dam), the second column dam (2$^\#$ dam), Yuejin dam (3$^\#$ dam), Songlin dam (4$^\#$ dam), battle dam (5$^\#$ dam), initial dam (6$^\#$ dam), Nanpo dam (7$^\#$ dam), Nanpoheng dam (8$^\#$ dam), Leigongtang dam (9$^\#$ dam) and east test dam (10$^\#$ dam). Owing to the long construction time, the large amount of engineering work and the limitation of productivity and technology at that time, the stability of the tailings pond has changed a lot. In addition, according to the seismic intensity zoning map of China, the area in which the tailings pond is located belongs to an earthquake area whose magnitude is less than six. In order to ensure the long-term stability and safety of the tailings pond, it is necessary to evaluate the stability and safety of all 10 dams, so as to provide guidance for the future decommissioning.

## 3.2. Dam stability evaluation index system

As there are numerous and complex factors to evaluate the stability of dams, this paper establishes the dam stability evaluation index system according to the relevant literature [1,6,10], combined with relevant Chinese standards [37,38] and the engineering practice of uranium tailings reservoirs studied. As shown in figure 2, this evaluation index system includes eight indicators, tailings accumulation dam crest elevation $X_1$, dam slope $X_2$, mechanical properties $X_3$, seepage capacity $X_4$, topographical conditions $X_5$, flood control capacity $X_6$, rainstorm resistance capacity $X_7$, and earthquake resistance capacity $X_8$. The specific values of each indicator are shown in table 1, and the value principles of each indicator are as follows.

Based on the field investigation, the index value of accumulation dam crest elevation $X_1$, dam slope $X_2$, mechanical property $X_3$, seepage capacity $X_4$ and earthquake resistance capacity $X_8$ are obtained. Among them, are many physical and mechanical properties of the dam. In this paper, 'shear strength', which is representative of the stability of the dam with strong correlation and relatively complete index statistics, is selected as the representative; 'seepage coefficient' represents the seepage capacity of the dam. Owing to the cumulative effect of external forces, the dam is prone to generating loose structures with thixotropic and liquefaction. When the precipitation and pore water pressure are too large, the mechanical properties will be reduced and the dam will be unstable. It is easy to ignore this in practical engineering applications. The larger the value is, the stronger the seepage capacity is, and the more unfavourable the stability of the dam is. The seismic capacity of the dam is expressed by the seismic coefficient calculated from the relevant report. The seismic coefficient is a safety factor derived

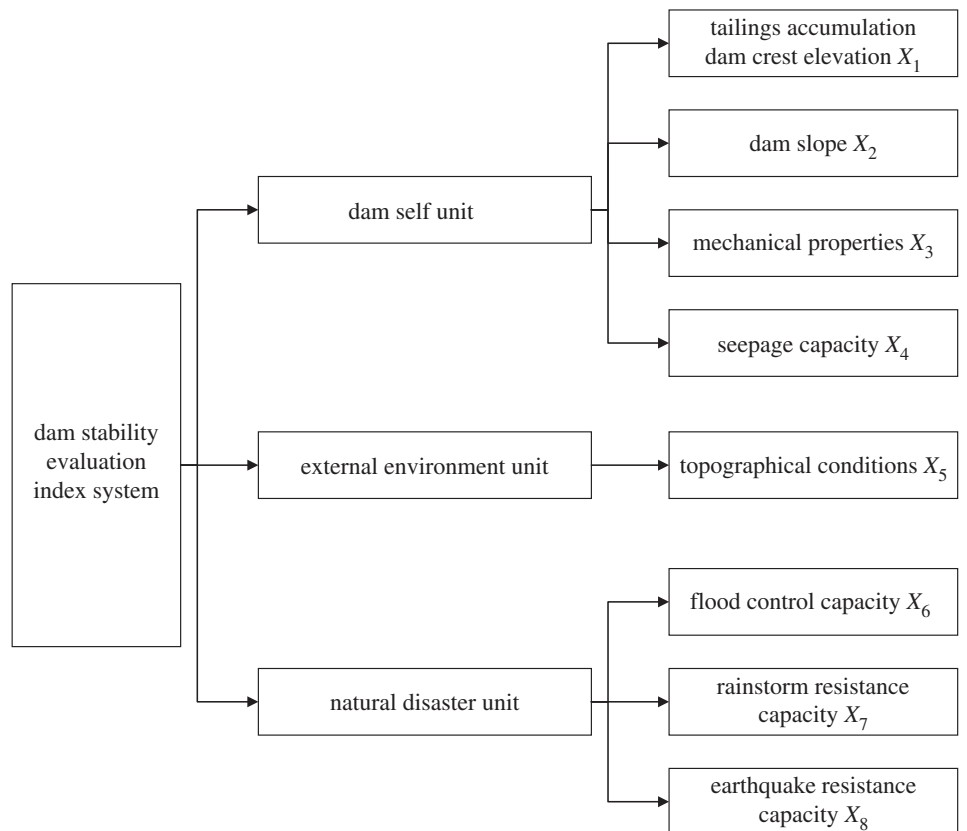

**Figure 2.** Dam stability evaluation system.

from the comprehensive consideration of gravity, water content, infiltration line, pore water pressure, and slippage of the sliding surface when the dam experiences a magnitude seven earthquake. The data, measured and calculated by engineers in the uranium tailings pond, could characterize the seismic capability of the dam.

Terrain condition $X_5$ is a qualitative index, which is divided into five grades: stable, relatively stable, general, relatively unstable and extremely unstable. And the corresponding scores are 10, 8, 6, 4 and 2, respectively.

Flood control resistance $X_6$ is a qualitative index, which is divided into five grades: strong, relatively strong, average, poor and range, and corresponding to the scores: 10, 8, 6, 4 and 2, respectively.

## 3.3. Computational results

According to table 1, the evaluation matrix is established:

$$
A = \begin{bmatrix}
93.9 & 0.248 & 65 & 9.6 \times 10^{-7} & 6 & 8 & 8 & 1.480 \\
84.5 & 0.299 & 52 & 3.2 \times 10^{-6} & 8 & 6 & 4 & 1.677 \\
83 & 0.326 & 13 & 2.6 \times 10^{-6} & 6 & 6 & 8 & 2.043 \\
98 & 0.441 & 42 & 3.5 \times 10^{-7} & 8 & 6 & 4 & 1.829 \\
90 & 0.667 & 23.4 & 3.78 \times 10^{-6} & 6 & 6 & 6 & 1.407 \\
97.5 & 0.763 & 25.3 & 2.66 \times 10^{-5} & 8 & 8 & 6 & 1.442 \\
87.5 & 0.667 & 30 & 2.85 \times 10^{-6} & 6 & 6 & 8 & 1.212 \\
87.5 & 0.667 & 30 & 2.85 \times 10^{-6} & 8 & 8 & 8 & 1.834 \\
95.3 & 0.4 & 19 & 2.2 \times 10^{-4} & 8 & 8 & 6 & 1.489 \\
80 & 0.4 & 40 & 2.4 \times 10^{-6} & 8 & 6 & 6 & 1.985
\end{bmatrix}.
$$

According to formulae (2.1)–(2.8), the fitting degree of each evaluation object and ideal solution can be obtained. The results are as follows.

As is calculated above, it can be concluded that the degree of conformity between the dam stability and the ideal solution is 76%, 93%, 82%, 90%, 66%, 79%, 85%, 96%, 32% and 89% successively. Among

**Table 1.** Evaluation index and value of dam stability. (Rainstorm resistance $X_7$ is a qualitative index, which is divided into five grades, namely: strong, relatively strong, average, poor and range. Corresponding scores are 10, 8, 6, 4 and 2 respectively.)

| dam number | accumulation dam crest elevation (m) $X_1$ | dam slope $X_2$ | mechanical property (KPa) $X_3$ | seepage capacity (cm s$^{-1}$) $X_4$ | topographical conditions $X_5$ | flood control resistance $X_6$ | rainstorm resistance capacity $X_7$ | seismic capacity $X_8$ |
|---|---|---|---|---|---|---|---|---|
| 1# | 93.9 | 1:4.03 | 65 | $9.6 \times 10^{-7}$ | 8 | 8 | 6 | 1.48 |
| 2# | 84.5 | 1:3.35 | 52 | $3.2 \times 10^{-6}$ | 8 | 6 | 4 | 1.677 |
| 3# | 83 | 1:3.07 | 13 | $2.6 \times 10^{-6}$ | 6 | 6 | 8 | 2.043 |
| 4# | 98 | 1:2.27 | 42 | $3.5 \times 10^{-7}$ | 8 | 6 | 4 | 1.829 |
| 5# | 90 | 1:1.50 | 23.4 | $3.78 \times 10^{-6}$ | 6 | 6 | 6 | 1.407 |
| 6# | 97.5 | 1:1.31 | 25.3 | $2.66 \times 10^{-5}$ | 8 | 8 | 6 | 1.442 |
| 7# | 87.5 | 1:1.50 | 30 | $2.85 \times 10^{-6}$ | 6 | 6 | 8 | 1.212 |
| 8# | 87.5 | 1:1.50 | 30 | $2.85 \times 10^{-6}$ | 8 | 8 | 8 | 1.834 |
| 9# | 95.3 | 1:2.50 | 19 | $2.2 \times 10^{-4}$ | 8 | 8 | 6 | 1.489 |
| 10# | 80 | 1:2.50 | 40 | $2.4 \times 10^{-6}$ | 8 | 6 | 6 | 1.985 |

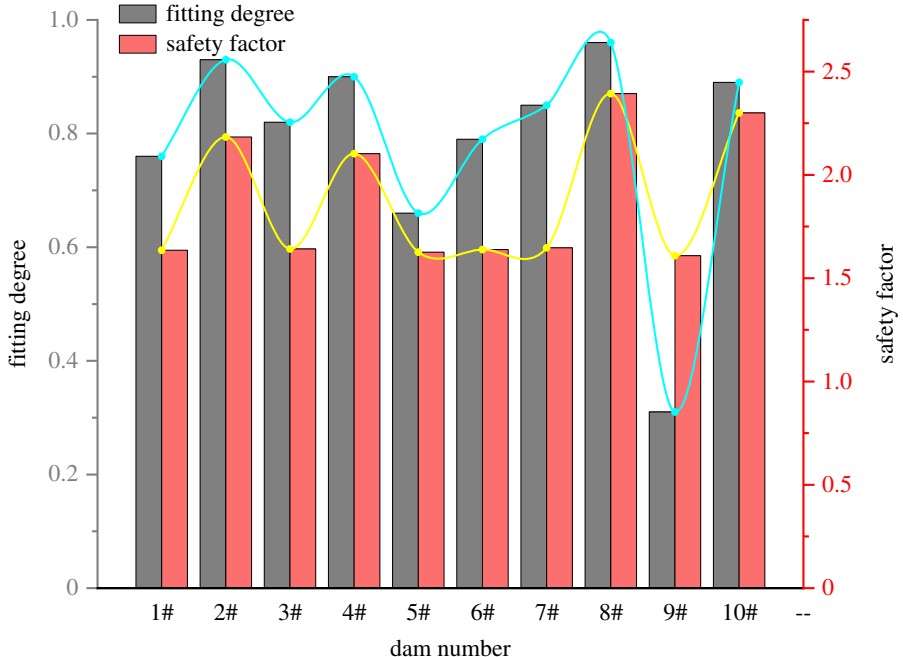

**Figure 3.** Comparison diagram of two evaluation methods.

**Table 2.** Fitting degree, safety factor and corresponding sequence.

| dam number | fitting degree | safety factor | dam fitting degree descending order | dam safety factor descending order |
|---|---|---|---|---|
| 1# | 0.76 | 1.635 | 8# | 8# |
| 2# | 0.93 | 2.183 | 2# | 10# |
| 3# | 0.82 | 1.641 | 4# | 2# |
| 4# | 0.90 | 2.102 | 10# | 4# |
| 5# | 0.66 | 1.626 | 7# | 7# |
| 6# | 0.79 | 1.638 | 3# | 3# |
| 7# | 0.85 | 1.647 | 6# | 6# |
| 8# | 0.96 | 2.393 | 1# | 1# |
| 9# | 0.31 | 1.609 | 5# | 5# |
| 10# | 0.89 | 2.3 | 9# | 9# |

the 10 dams of the tailings pond, the dam bodies with good stability are: 2# dam and 8# dam; dam with poor stability is: 9# dam.

# 4. Discussion

## 4.1. Comparison with the traditional safety coefficient method

On this basis, the fitting degree results obtained by the evaluation model are compared with the traditional safety coefficient which are calculated via the relevant data of the uranium tailings pond (table 2). In order to make a more intuitive comparison between the two, the fitting degree and the calculated value of the safety coefficient in the data are normalized, respectively, to obtain figure 3.

When sorting by safety factor, the dams at the end are: 9# dam and 5# dam. The actual results are in good agreement with the results calculated by the TOPSIS model under the applied variable weight theory. In addition, when the stability factor is evaluated with reference to the safety factor, the values

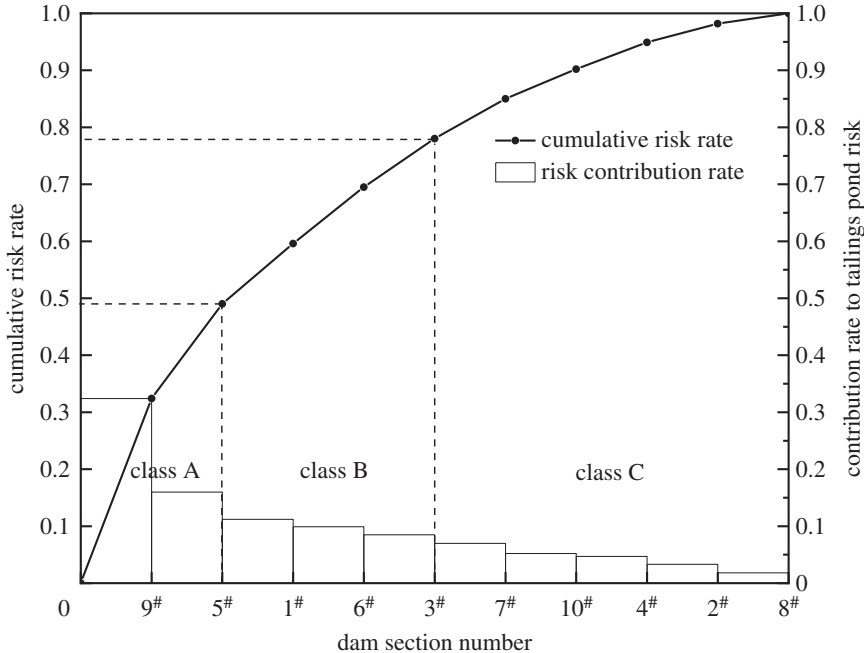

**Figure 4.** The contribution rate of each dam to the overall risk of tailings pond.

**Table 3.** ABC analysis classification.

| dam number | self-risk | contribution rate to the whole tailings pond risk | classification |
|---|---|---|---|
| 9# | 0.69 | 0.324 | A |
| 5# | 0.34 | 0.160 | A |
| 1# | 0.24 | 0.112 | B |
| 6# | 0.21 | 0.099 | B |
| 3# | 0.18 | 0.085 | B |
| 7# | 0.15 | 0.070 | C |
| 10# | 0.11 | 0.052 | C |
| 4# | 0.10 | 0.047 | C |
| 2# | 0.07 | 0.033 | C |
| 8# | 0.04 | 0.018 | C |

of some dams are relatively close, which is not conducive to the effective stability of the dam body. It can be seen from the above chart that the differences in safety factor values for 1# dam, 3# dam, 5# dam, 6# dam, 7# dam and 9# dam are very small, which makes it difficult for decision makers to select the needs of several similar evaluation results. The dams that are of primary concern cannot provide accurate guidance for the decommissioning of tailings ponds. Under variable weight theory, the TOPSIS model can effectively magnify the difference between dam bodies with poor stability by using 'negative value' and 'penalty quantity', which is convenient to draw the evaluation conclusion. The uranium tailings reservoir evaluated in this paper is in the decommissioning and treatment stage. Hence, the condition of the dam will change accordingly. According to the latest monitoring data of the uranium tailings dam, the evaluation results are still consistent with the results of this paper.

## 4.2. Analysis by using activity-based classification (ABC analysis method)

The risk degree of each dam can be obtained from the corresponding stability fitting degree in table 2, and then the rate of risk contributed by each dam to the overall risk of the tailings pond (the ratio of the instability risk rate of each dam to the sum of all dam body risk rates) is calculated (figure 4). The dams are classified to guide safety inputs and decommissioning, as shown in table 3.

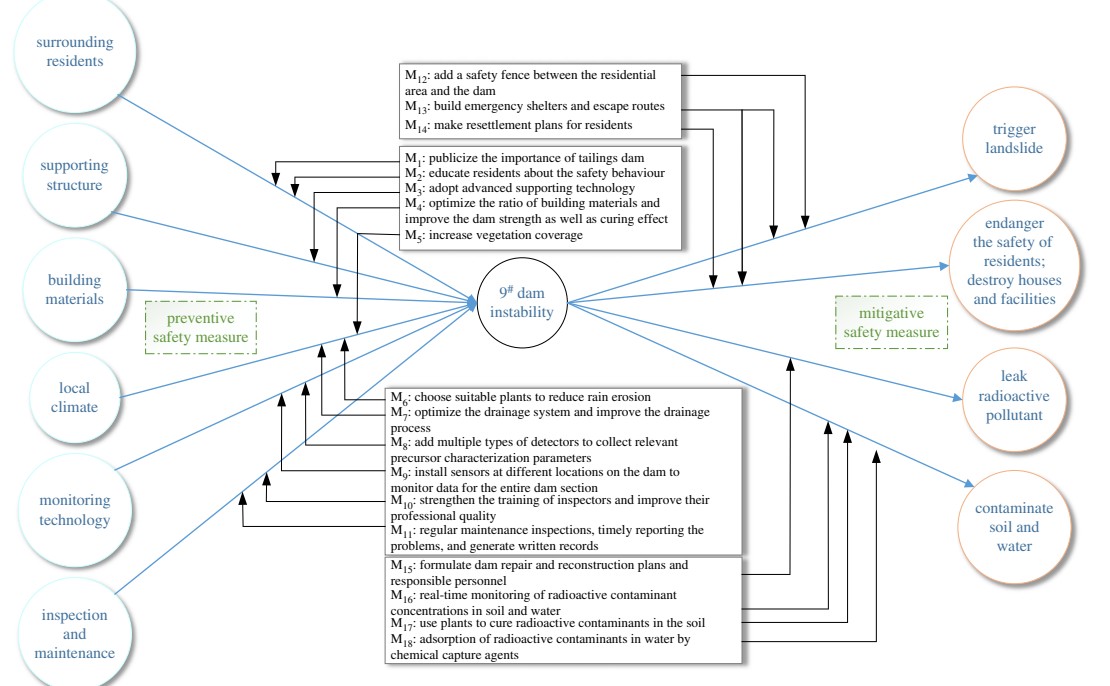

**Figure 5.** Bow tie analysis of 9# dam instability.

It can be seen from figure 3 that the contribution of the 9# dam and the 5# dam to the overall risk of the tailings pond is nearly 50%, which is close to the sum of the other eight dams. Therefore, they are classified as class A, which is the key minority that plays a decisive role in the overall risk of the tailings pond, and should be given the greatest attention and safety investment in the subsequent management; 1# dam, 6# dam and 3# dam are classified as class B, which have a secondary impact on the overall risk of the tailings pond. Measures should be taken in prevention and control work; 7# dam, 10# dam, 4# dam, 2# dam and 8# dam do not have obvious adverse effects on the overall stability of the tailings pond, and are classified as class C. It is necessary to regularly inspect and monitor relevant indicators and parameters.

## 4.3. Bow tie analysis of the critical event

It can be seen from the above analysis that the 9# dam has the highest probability of instability and the greatest impact on the overall safety of the uranium tailings pond. In previous studies, the safety assessment of dam stability was only to identify weak links through simple analysis, which was difficult when drawing the cause and possible consequences of the critical event, and it was hard to give effective suggestions to improve the safety situation of the tailings dam [39,40]. Based on this, the instability of 9# dam was analysed as a critical event by the bow tie method (figure 5), and its causes and consequences of which were identified and preventive measures were given.

The left side of the bow tie model includes six causes of 9# dam instability and 11 preventive safety measures. The right side contains the potential consequences and seven mitigative safety measures. Through the analysis of the bow tie method, the control idea for the dam instability will be clearer and more organized, which is helpful to reduce the possibility of dam instability.

## 4.4. Brief summary of discussion

This paper successfully applied the TOPSIS model based on variable weight theory to the stability evaluation of a uranium tailings pond. On the basis of ensuring the accuracy of the results, this model can identify the weak link (critical event) that affects the stability of the tailings dam group and provide a strong basis for subsequent risk analysis and accident prevention. In order to further

clarify the impact of dam instability on the overall stability of the tailings pond, this paper introduces the ABC analysis method, the classic method in the field of economics, to characterize the contribution rate of each dam to the overall risk of the tailings pond. The analysis can help managers to understand the stability of the dam more clearly and intuitively, so as to develop safety measures in a pertinent manner. In order to make more comprehensive and systematic safety measures, the bow tie model is adopted in this paper to analyse the instability of $9^{\#}$ dam from the causes and consequences of the accident, and 11 preventive safety measures and seven mitigative safety measures are obtained.

The application of the ABC analysis method and bow tie model to the risk assessment of uranium tailings dam is rare in previous studies and can be popularized in relevant safety assessment issues. In addition, for the convenience of discussion, this paper only analyses the $9^{\#}$ dam with the greatest possibility of instability by the bow tie model. In future studies, more dams should be taken as a critical event to analyse via the bow tie model, so as to obtain the commonness and differences of different dams in the safety measures, providing a theoretical basis for decommissioning and radiation protection.

# 5. Conclusion

(i) The variable weight theory is introduced into the stability evaluation of each dam section of the uranium tailings reservoir, which can effectively correct the constant weight and make the weight value more close to the actual situation. In addition, combining the variable weight theory with the TOPSIS method can provide a scientific and reliable new idea for the safety of tailings ponds and other multi-index comprehensive evaluations.

(ii) For tailings ponds, especially uranium tailings ponds, the safety of dams is very important. In this paper, the TOPSIS model under variable weight theory and the basic method of mathematical analysis are used to evaluate the dam stability of a tailings reservoir, and the result obtained is highly consistent with the actual calculation results. According to the calculation, the fitting degree of dam stability and ideal solution of the first column dam, the second column dam, Yuejin dam, Songlin dam, battle dam, initial dam, Nanpo dam, Nanpoheng dam, the Leigongtang dam and the east test dam is 76%, 93%, 82%, 90%, 66%, 79%, 85%, 96%, 32% and 89%, successively. Namely: among the 10 dams in a uranium tailings pond in the south, the stability of the combat dam and the Leigongtang dam are the worst, and it is necessary to take measures to ensure their safety and reliability in the subsequent decommissioning work.

(iii) This paper verifies that the TOPSIS model based on variable weight theory can solve the problem of multi-factor evaluation in engineering practice, and the idea of equilibrium is conducive to magnifying the subtle differences between evaluation objects, drawing evaluation conclusions, and avoiding decision-making errors caused by inconsistencies between decision-making and practice.

(iv) Based on systematic thinking, the bow tie analysis method establishes safety barriers from the two aspects of accident cause and accident consequence to block the path of accident occurrence, which is conducive to a deeper understanding of the critical event. The safety measures obtained by the bow tie model can prevent accidents more effectively. In this paper, the ABC analysis method and the bow tie model are combined to provide a new idea for risk identification and risk control of a uranium tailings pond, making the evaluation system more complete.

# Appendix

$$
A = \begin{bmatrix}
93.9 & 0.248 & 65 & 9.6 \times 10^{-7} & 6 & 8 & 8 & 1.480 \\
84.5 & 0.299 & 52 & 3.2 \times 10^{-6} & 8 & 6 & 4 & 1.677 \\
83 & 0.326 & 13 & 2.6 \times 10^{-6} & 6 & 6 & 8 & 2.043 \\
98 & 0.441 & 42 & 3.5 \times 10^{-7} & 8 & 6 & 4 & 1.829 \\
90 & 0.667 & 23.4 & 3.78 \times 10^{-6} & 6 & 6 & 6 & 1.407 \\
97.5 & 0.763 & 25.3 & 2.66 \times 10^{-5} & 8 & 8 & 6 & 1.442 \\
87.5 & 0.667 & 30 & 2.85 \times 10^{-6} & 6 & 6 & 8 & 1.212 \\
87.5 & 0.667 & 30 & 2.85 \times 10^{-6} & 8 & 8 & 8 & 1.834 \\
95.3 & 0.4 & 19 & 2.2 \times 10^{-4} & 8 & 8 & 6 & 1.489 \\
80 & 0.4 & 40 & 2.4 \times 10^{-6} & 8 & 6 & 6 & 1.985
\end{bmatrix}.
$$

The multi-attribute evaluation matrix can be transformed into normalized forms:

$$X = \begin{bmatrix} 0.104 & 0.051 & 0.191 & 0.00361 & 0.0835 & 0.118 & 0.125 & 0.090 \\ 0.094 & 0.061 & 0.153 & 0.01201 & 0.111 & 0.088 & 0.064 & 0.102 \\ 0.093 & 0.067 & 0.038 & 0.00979 & 0.0835 & 0.088 & 0.125 & 0.125 \\ 0.109 & 0.090 & 0.124 & 0.00132 & 0.111 & 0.088 & 0.064 & 0.112 \\ 0.100 & 0.137 & 0.069 & 0.01431 & 0.0835 & 0.088 & 0.093 & 0.086 \\ 0.109 & 0.156 & 0.075 & 0.10015 & 0.111 & 0.118 & 0.093 & 0.088 \\ 0.098 & 0.137 & 0.088 & 0.01073 & 0.0835 & 0.088 & 0.125 & 0.074 \\ 0.098 & 0.137 & 0.088 & 0.01073 & 0.111 & 0.118 & 0.125 & 0.112 \\ 0.106 & 0.082 & 0.056 & 0.82831 & 0.111 & 0.118 & 0.093 & 0.090 \\ 0.089 & 0.082 & 0.118 & 0.00904 & 0.111 & 0.088 & 0.093 & 0.121 \end{bmatrix}.$$

Constant weight value of each evaluation index is as follows:

$$W = (\,0.089 \quad 0.126 \quad 0.273 \quad 0.273 \quad 0.034 \quad 0.051 \quad 0.076 \quad 0.078\,),$$

according to the constant weight vector and formula (2.2) in the main text, we calculated the state variable weight vector:

$$S = \begin{bmatrix} 1 & 1.025 & 1 & 1.049 & 1.008 & 1 & 1 & 1.005 \\ 1.003 & 1.020 & 1 & 1.045 & 1 & 1.006 & 1.018 & 1 \\ 1.004 & 1.017 & 1.031 & 1.046 & 1.008 & 1.006 & 1 & 1 \\ 1 & 1.005 & 1 & 1.051 & 1 & 1.006 & 1.018 & 1 \\ 1 & 1 & 1.016 & 1.044 & 1.008 & 1.006 & 1.004 & 1.007 \\ 1 & 1 & 1.013 & 1 & 1 & 1 & 1.004 & 1.006 \\ 1.001 & 1 & 1.006 & 1.046 & 1.008 & 1.006 & 1 & 1.013 \\ 1.001 & 1 & 1.006 & 1.046 & 1 & 1 & 1 & 1 \\ 1 & 1.009 & 1.022 & 0.695 & 1 & 1 & 1.004 & 1.005 \\ 1.006 & 1.009 & 1 & 1.047 & 1 & 1.006 & 1.004 & 1 \end{bmatrix}.$$

(1) Calculating the Hardarmard product:

$$W \cdot S = \begin{bmatrix} 0.0890 & 0.1292 & 0.2730 & 0.2864 & 0.0343 & 0.0510 & 0.0760 & 0.0784 \\ 0.0893 & 0.1285 & 0.2730 & 0.2853 & 0.0340 & 0.0513 & 0.0774 & 0.0780 \\ 0.0894 & 0.1281 & 0.2815 & 0.2856 & 0.0343 & 0.0513 & 0.0760 & 0.0780 \\ 0.0890 & 0.1266 & 0.2730 & 0.2869 & 0.0340 & 0.0513 & 0.0774 & 0.0780 \\ 0.0890 & 0.1260 & 0.2774 & 0.2850 & 0.0343 & 0.0513 & 0.0763 & 0.0785 \\ 0.0890 & 0.1260 & 0.2765 & 0.2730 & 0.0340 & 0.0510 & 0.0763 & 0.0785 \\ 0.0891 & 0.1260 & 0.2746 & 0.2856 & 0.0343 & 0.0513 & 0.0760 & 0.0790 \\ 0.0891 & 0.1260 & 0.2746 & 0.2856 & 0.0340 & 0.0510 & 0.0760 & 0.0780 \\ 0.0890 & 0.1271 & 0.2820 & 0.1897 & 0.0340 & 0.0510 & 0.0763 & 0.0784 \\ 0.0895 & 0.1271 & 0.2730 & 0.2858 & 0.0340 & 0.0513 & 0.0763 & 0.0780 \end{bmatrix}.$$

(2) Calculating the variable weight vector matrix:

$$W(X) = \begin{bmatrix} 0.0875 & 0.0127 & 0.2684 & 0.2815 & 0.0337 & 0.0501 & 0.0747 & 0.0771 \\ 0.0878 & 0.1264 & 0.2685 & 0.2806 & 0.0334 & 0.0504 & 0.0761 & 0.0767 \\ 0.0873 & 0.1251 & 0.2748 & 0.2789 & 0.0335 & 0.0501 & 0.0742 & 0.0762 \\ 0.0876 & 0.1246 & 0.2686 & 0.2824 & 0.0335 & 0.0505 & 0.0762 & 0.0768 \\ 0.0874 & 0.0124 & 0.2725 & 0.2800 & 0.0337 & 0.0504 & 0.0750 & 0.0771 \\ 0.0886 & 0.1255 & 0.2752 & 0.2717 & 0.0341 & 0.0508 & 0.0760 & 0.0781 \\ 0.0877 & 0.1240 & 0.2703 & 0.2811 & 0.0338 & 0.0505 & 0.0748 & 0.0778 \\ 0.0878 & 0.1242 & 0.2707 & 0.2816 & 0.0335 & 0.0502 & 0.0749 & 0.0769 \\ 0.0960 & 0.1370 & 0.3040 & 0.2045 & 0.0367 & 0.0549 & 0.0823 & 0.0845 \\ 0.0882 & 0.1252 & 0.2690 & 0.2816 & 0.0335 & 0.0505 & 0.0752 & 0.0768 \end{bmatrix}.$$

Based on $X_1$, $X_2$, $X_4$ being negative indicators and $X_3$, $X_5$, $X_6$, $X_7$, $X_8$ being positive indicators, the positive and negative ideal solutions are calculated, which combined with formulae (2.4) and (2.5) are as follows:

$$\begin{cases} C^+ = [0.0078 \quad 0.00065 \quad 0.051 \quad 0.0037 \quad 0.0041 \quad 0.0065 \quad 0.0094 \quad 0.0095] \\ C^- = [0.0870 \quad 0.02000 \quad 0.010 \quad 0.1700 \quad 0.0028 \quad 0.0044 \quad 0.0049 \quad 0.0058] \end{cases}.$$

The distance between the evaluation object and the ideal solution is calculated by formulae (2.6) and (2.7), and the results are as follows:

$$\begin{cases} d_1^+ = 0.0042 \\ d_1^- = 0.0133 \end{cases} \begin{cases} d_2^+ = 0.0133 \\ d_2^- = 0.1873 \end{cases} \begin{cases} d_3^+ = 0.0418 \\ d_3^- = 0.1854 \end{cases} \begin{cases} d_4^+ = 0.0217 \\ d_4^- = 0.1881 \end{cases} \begin{cases} d_5^+ = 0.0855 \\ d_5^- = 0.1673 \end{cases}$$
$$\begin{cases} d_6^+ = 0.0428 \\ d_6^- = 0.1630 \end{cases} \begin{cases} d_7^+ = 0.0519 \\ d_7^- = 0.1251 \end{cases} \begin{cases} d_8^+ = 0.0316 \\ d_8^- = 0.8776 \end{cases} \begin{cases} d_9^+ = 0.1701 \\ d_9^- = 0.0771 \end{cases} \begin{cases} d_{10}^+ = 0.0205 \\ d_{10}^- = 0.1969 \end{cases}.$$

Thus, the fitting degree of each evaluation object and ideal solution can be obtained. The results are as follows:

$$E_1^+ = 0.76,\ E_2^+ = 0.93,\ E_3^+ = 0.82,\ E_4^+ = 0.90,\ E_5^+ = 0.66,$$
$$E_6^+ = 0.79,\ E_7^+ = 0.85,\ E_8^+ = 0.96,\ E_9^+ = 0.31,\ E_{10}^+ = 0.89.$$

Data accessibility. According to table 1 in the main text, the multi-attribute evaluation matrix is established.

Authors' contributions. J.F. developed the research plan, guided the entire process of writing, carried out the final approval of the version to be published; W.H. carried out drafting of the article or revising it critically for important intellectual content and agree to be accountable for all aspects of the work in ensuring that questions related to the accuracy or integrity of any part of the work are appropriately investigated and resolved; L.Y. participated in the design of the study. C.G. coordinated the study and helped draft the manuscript; G.J. carried out the substantial contributions to analysis and interpretation of data; W.Z. carried out the preliminary preparation and collected field data. All authors gave final approval for publication.

Competing interests. We declare we have no competing interests.

Funding. This research was supported by the National Natural Science Foundation of China (grant nos. 11475081 and 11875164), project approved by the Research Foundation of Education Bureau of Hunan Province, China (grant no. 17B228), the Double First Class Construct Program of USC (grant no. 2017SLY05), Hunan Provincial Innovation Foundation For Postgraduate, China (grant no. CX20190726), project approved by Hunan Province Engineering Research Center of Radioactive Control Technology in Uranium Mining and Metallurgy and Hunan Province Engineering Technology Research Center of Uranium Tailings Treatment Technology (grant no. 2018YKZX1004), Open Fund Project of Hunan Cooperative Innovation Center for Nuclear Fuel Cycle Technology and Equipment (grant no. 2019KFZ01).

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
