## [Reviewer comments · Royal Society Open Science]

Review History

RSOS-191566.R0 (Original submission)

Review form: Reviewer 1

Is the manuscript scientifically sound in its present form?

Yes

Are the interpretations and conclusions justified by the results?

Yes

Is the language acceptable?

Yes

Do you have any ethical concerns with this paper?

No

Have you any concerns about statistical analyses in this paper?

No

Recommendation?

Accept with minor revision (please list in comments)

Comments to the Author(s)

Dear editor,

I have review the manuscript entitled “Comprehensive evaluation system for stability of multiple dams in uranium tailings reservoir: based on TOPSIS model and bow tie model”, which concerns to solve the problem of safety evaluation of tailings dam stability by introducing a combined TOPSIS and bow tie model. The manuscript is in general written in good style and the results are reasonable. However, there are some comments which presented below:

- 1- There are some misunderstanding and grammatical errors that should be corrected.
- 2- The explanation and figure regarding bow tie diagram is poor and it is better to rich them.
- 3- Have the dam stability evaluation system been developed by the authors?
- 4- How the seismic capacity was calculated?
- 5- Why the safety factor for dam #9 is less than 1? Although it is analyzed in bow tie diagram (Fig. 5).
- 6- The contribution risk rate was not clearly defined.
- 7- It is better if the authors compare their results with more recently findings in discussion section.

Regards,

Review form: Reviewer 2

Is the manuscript scientifically sound in its present form?

Yes

Are the interpretations and conclusions justified by the results?

Yes

Is the language acceptable?

Yes

Do you have any ethical concerns with this paper?

No

Have you any concerns about statistical analyses in this paper?

No

Recommendation?

Accept with minor revision (please list in comments)

Comments to the Author(s)

The paper proposes TOPSIS model and bow tie for safety evaluation of tailings dam stability. The paper can be published after minor revision as given in following.

1. The rest of the paper should be added at the end of the introduction.
2. In literature, some safety evaluation paper on dam is missing. The literature review can be enriched by adding different case study considering safety evaluation in dam.
3. Why did you use TOPSIS instead of other MCDM such as VIKOR, PROMETHEE, ELECTRE etc. Please explain.

Review form: Reviewer 3

Is the manuscript scientifically sound in its present form?

Yes

Are the interpretations and conclusions justified by the results?

Yes

Is the language acceptable?

Yes

Do you have any ethical concerns with this paper?

No

Have you any concerns about statistical analyses in this paper?

No

Recommendation?

Reject

Comments to the Author(s)

The manuscript does not provide any new information about safety of dams. This is just an application of some search methods with some data. Moreover the applied metric for dam stability is very simplistic.

Decision letter (RSOS-191566.R0)

03-Jan-2020

Dear Dr Wu,

The editors assigned to your paper ("Comprehensive evaluation system for stability of multiple dams in uranium tailings reservoir: based on TOPSIS model and bow tie model") have now received comments from reviewers. We would like you to revise your paper in accordance with the referee and Associate Editor suggestions which can be found below (not including confidential reports to the Editor). Please note this decision does not guarantee eventual acceptance.

Please submit a copy of your revised paper before 26-Jan-2020. Please note that the revision deadline will expire at 00.00am on this date. If we do not hear from you within this time then it will be assumed that the paper has been withdrawn. In exceptional circumstances, extensions may be possible if agreed with the Editorial Office in advance. We do not allow multiple rounds of revision so we urge you to make every effort to fully address all of the comments at this stage. If deemed necessary by the Editors, your manuscript will be sent back to one or more of the original reviewers for assessment. If the original reviewers are not available, we may invite new reviewers.

- Data accessibility

If you wish to submit your supporting data or code to Dryad (<http://datadryad.org/>), or modify your current submission to dryad, please use the following link:
<http://datadryad.org/submit?journalID=RSOS&manu=RSOS-191566>

- Competing interests

- Authors' contributions

- Acknowledgements

- Funding statement

on behalf of Prof R. Kerry Rowe (Subject Editor)
 openscience@royalsociety.org

Associate Editor's comments:

The Editors are concerned that the paper in its current form is missing a number of key aspects, as identified by the reviewers. In its current form, the paper is not publishable. We're also concerned by the comments of the third reviewer that the manuscript does not represent a meaningful contribution or advance on the existing body of literature.

Your revision must not only include changes to respond to the concerns raised by the first two reviewers but also provide a thorough explanation (in both the manuscript itself and your response to reviewers document) of how your paper moves the field forward. If you are unable to demonstrate this to the Editors and the reviewers, we regret the revision will not be published.

Comments to Author:

Reviewers' Comments to Author:

Reviewer: 1

Comments to the Author(s)

Dear editor,

I have review the manuscript entitled "Comprehensive evaluation system for stability of multiple dams in uranium tailings reservoir: based on TOPSIS model and bow tie model", which concerns to solve the problem of safety evaluation of tailings dam stability by introducing a combined TOPSIS and bow tie model. The manuscript is in general written in good style and the results are reasonable. However, there are some comments which presented below:

- 1- There are some misunderstanding and grammatical errors that should be corrected.
- 2- The explanation and figure regarding bow tie diagram is poor and it is better to rich them.
- 3- Have the dam stability evaluation system been developed by the authors?
- 4- How the seismic capacity was calculated?
- 5- Why the safety factor for dam #9 is less than 1? Although it is analyzed in bow tie diagram (Fig. 5).
- 6- The contribution risk rate was not clearly defined.
- 7- It is better if the authors compare their results with more recently findings in discussion section.

Regards,

Reviewer: 2

Comments to the Author(s)

The paper proposes TOPSIS model and bow tie for safety evaluation of tailings dam stability. The paper can be published after minor revision as given in following.

1. The rest of the paper should be added at the end of the introduction.
2. In literature, some safety evaluation paper on dam is missing. The literature review can be enriched by adding different case study considering safety evaluation in dam.
3. Why did you use TOPSIS instead of other MCDM such as VIKOR, PROMETHEE, ELECTRE etc. Please explain.

Reviewer: 3

Comments to the Author(s)

The manuscript does not provide any new information about safety of dams.
This is just an application of some search methods with some data.
Moreover the applied metric for dam stability is very simplistic.

Author's Response to Decision Letter for (RSOS-191566.R0)

See Appendix A.

RSOS-191566.R1 (Revision)

Review form: Reviewer 3

Is the manuscript scientifically sound in its present form?

Yes

Are the interpretations and conclusions justified by the results?

Yes

Is the language acceptable?

Yes

Do you have any ethical concerns with this paper?

No

Have you any concerns about statistical analyses in this paper?

No

Recommendation?

Accept as is

Comments to the Author(s)

--

Review form: Reviewer 4

Is the manuscript scientifically sound in its present form?

Yes

Are the interpretations and conclusions justified by the results?

Yes

Is the language acceptable?

Yes

Do you have any ethical concerns with this paper?

No

Have you any concerns about statistical analyses in this paper?

No

Recommendation?

Accept with minor revision (please list in comments)

Comments to the Author(s)

The abstract should be corrected to:

"The main purpose of this study is to analyse the evaluation of tailings dam stability safety under multiple...."

Review form: Reviewer 5 (Giyasuddin Siddique)

Is the manuscript scientifically sound in its present form?

Yes

Are the interpretations and conclusions justified by the results?

Yes

Is the language acceptable?

No

Do you have any ethical concerns with this paper?

No

Have you any concerns about statistical analyses in this paper?

No

Recommendation?

Accept with minor revision (please list in comments)

Comments to the Author(s)

I have come across the whole paper including every sentences, lines, words, diagrams and calculations. The relevance of the paper and its standard is quite satisfactory. I have also come across every sentences changed and replaced by the previous reviewers. But I must say that the English of the paper is poor, for which some of the sentences are to be reconstructed. However my humble suggestions and corrections includes the following -

1. Page 6, Line 28: Replace "in" With "of".

2. Page 6, Line 31: Causes Please Check the Grammatical Errors.
3. Page 7, Line 43: Belongs to.
4. Page 7, Line 44: After Semicolon Start with Small Letter.
5. Page 7, Line 82: Check Spelling.
6. Page 8, Line 118: "Indices" Make Correction Elsewhere.
7. Page 9, Line 146: Small Letter "c".
8. Page 9, Line 159: Poor Sentence Construction.
9. Page 12, Line 237: Is "Consequent Upon" Instead of Related To.
10. Page 12, Line 238: It Should Be "Sustainability" Instead Of Sustainable Development.
11. Page 13, Line 271: Change The Sentence, It Is Poorly Constructed.
12. Page 16, Line 319: Mention The Values Accepted In The Chart To Mark High Or Low.
13. Page 16, Line 327: Provide Citation.
14. Page 16, Line 331: "The Rate of Risk Contributed By Each Dam".
15. Page 17, Line 352: Give Citation.
16. Page 19, Line 396: Instead of Evaluation Conclusion Write "Result Obtained".

Decision letter (RSOS-191566.R1)

26-Feb-2020

Dear Dr Jiang:

On behalf of the Editors, I am pleased to inform you that your Manuscript RSOS-191566.R1 entitled "Comprehensive evaluation system for stability of multiple dams in uranium tailings reservoir: based on TOPSIS model and bow tie model" has been accepted for publication in Royal Society Open Science subject to minor revision in accordance with the referee suggestions. Please find the referees' comments at the end of this email.

The reviewers and Subject Editor have recommended publication, but also suggest some minor revisions to your manuscript. Therefore, I invite you to respond to the comments and revise your manuscript.

- Ethics statement

- Data accessibility

<http://datadryad.org/submit?journalID=RSOS&manu=RSOS-191566.R1>

- **Competing interests**

- **Authors' contributions**

- **Acknowledgements**

- **Funding statement**

Because the schedule for publication is very tight, it is a condition of publication that you submit the revised version of your manuscript before 06-Mar-2020. Please note that the revision deadline will expire at 00.00am on this date. If you do not think you will be able to meet this date please let me know immediately.

1) A text file of the manuscript (tex, txt, rtf, docx or doc), references, tables (including captions) and figure captions. Do not upload a PDF as your "Main Document".

- 2) A separate electronic file of each figure (EPS or print-quality PDF preferred (either format should be produced directly from original creation package), or original software format)
- 3) Included a 100 word media summary of your paper when requested at submission. Please ensure you have entered correct contact details (email, institution and telephone) in your user account
- 4) Included the raw data to support the claims made in your paper. You can either include your data as electronic supplementary material or upload to a repository and include the relevant doi within your manuscript
- 5) All supplementary materials accompanying an accepted article will be treated as in their final form. Note that the Royal Society will neither edit nor typeset supplementary material and it will be hosted as provided. Please ensure that the supplementary material includes the paper details where possible (authors, article title, journal name).

on behalf of R. Kerry Rowe (Subject Editor)
openscience@royalsociety.org

Associate Editor Comments to Author:
Comments to the Author:

As you will see, three reviewers have provided commentary on your manuscript; however, the Editors have been forced to discount the comments of the first reviewer as, regrettably, they have not provided sufficient reasoning for their recommendation.

Fortunately, the other reviewers have provided somewhat more explanation for their recommendation. Based on the last reviewer's comments, we would ask that you seek professional language guidance before you submit your revision (<https://royalsociety.org/journals/authors/benefits/language-editing/>) - if you need a short extension to the revision deadline to facilitate this, please let the editorial office know at openscience@royalsociety.org, and they will be happy to assist. That said, acceptance of the manuscript is subject to you providing evidence that a professional language editing service has been used to support your submission - if you do not provide this evidence (a certificate of language editing, for example), we will not be able to accept the paper, and it will be returned to you.

Other than this requirement, please ensure you address the remaining comments from the referees carefully.

Reviewer comments to Author:

Reviewer: 4

Comments to the Author(s)

The abstract should be corrected to:

"The main purpose of this study is to analyse the evaluation of tailings dam stability safety under multiple....

Reviewer: 5

Comments to the Author(s)

I have come across the whole paper including every sentences, lines, words, diagrams and calculations. The relevance of the paper and its standard is quite satisfactory. I have also come across every sentences changed and replaced by the previous reviewers. But I must say that the English of the paper is poor, for which some of the sentences are to be reconstructed. However my humble suggestions and corrections includes the following -

1. Page 6, Line 28: Replace "in" With "of".
2. Page 6, Line 31: Causes Please Check the Grammatical Errors.
3. Page 7, Line 43: Belongs to.
4. Page 7, Line 44: After Semicolon Start with Small Letter.
5. Page 7, Line 82: Check Spelling.
6. Page 8, Line 118: "Indices" Make Correction Elsewhere.
7. Page 9, Line 146: Small Letter "c".
8. Page 9, Line 159: Poor Sentence Construction.
9. Page 12, Line 237: Is "Consequent Upon" Instead of Related To.
10. Page 12, Line 238: It Should Be "Sustainability" Instead Of Sustainable Development.
11. Page 13, Line 271: Change The Sentence, It Is Poorly Constructed.
12. Page 16, Line 319: Mention The Values Accepted In The Chart To Mark High Or Low.
13. Page 16, Line 327: Provide Citation.
14. Page 16, Line 331: "The Rate of Risk Contributed By Each Dam".
15. Page 17, Line 352: Give Citation.
16. Page 19, Line 396: Instead of Evaluation Conclusion Write "Result Obtained".

Author's Response to Decision Letter for (RSOS-191566.R1)

See Appendix B.

Decision letter (RSOS-191566.R2)

16-Mar-2020

Dear Dr Jiang,

It is a pleasure to accept your manuscript entitled "Comprehensive evaluation system for stability of multiple dams in uranium tailings reservoir: based on TOPSIS model and bow tie model" in its current form for publication in Royal Society Open Science. The comments of the reviewer(s) who reviewed your manuscript are included at the foot of this letter.

Please ensure that you send to the editorial office an editable version of your accepted

manuscript, and individual files for each figure and table included in your manuscript. You can send these in a zip folder if more convenient. Failure to provide these files may delay the processing of your proof. You may disregard this request if you have already provided these files to the editorial office.

Kind regards,

Anita Kristiansen
Editorial Coordinator

on behalf of R. Kerry Rowe (Subject Editor)
openscience@royalsociety.org

Appendix A

Dear Editors and Reviewers:

Thanks very much for taking your time to review this manuscript. I really appreciate all reviewers' comments and suggestions! All of suggestions have enabled us to improve our work. Based on the instructions provided in your letter, we uploaded the file of the revised manuscript and our responses are given in a different color (red). The following is the reply to the referees' comments.

For Reviewer 1:

Thank you for your efforts. I have replied to your suggestions point by point. The statements are as follow:

1. (1) Removed “with a huge amount of uranium tailings” at line 231;
 - (2) Modified “the stability of the tailings ponds is related to the safety of nearby residents and the Xiangjiang waters” to “the stability of the tailings ponds is related to nearby residents' safety and the Xiangjiang rivers' sustainable development.” at line 237-238;
 - (3) Modified “The total length of the 10 dam sections is 4600 meters” to “whose length is 4600 meters” at line 240;
 - (4) Modified “the area where the tailings pond is located belongs to an earthquake area less than 6 degree.” to “the area in which the tailings pond is located belongs to an earthquake area whose magnitude is less than six.” at line 248-250;
 - (5) Removed “will” at line 340;
 - (6) Modified “will” to “do” at line 342;
 - (7) Modified “will be” to “is” at line 343;
 - (8) Modified “thus reducing the possibility of dam instability” to “which is helpful to reduce the possibility of dam instability.” at line 359-360;
 - (9) Modified “analyzed” to “analyze” at line 379
2. The explanation and figure(Fig. 1) regarding bow tie diagram have been enriched as follow: “Safety barrier functions is to prevent, control, or mitigate undesired events or accidents. If a barrier function is performed successfully, it should have a direct and significant effect on the occurrence or consequences of an undesired accident[35]; Barrier system is a barrier group in which multiple safety barriers collectively implement or perform a barrier function, and it is inappropriate to include all barriers in a single level. Besides the title that is most often used (Escalation Factor) contributes to poorly understood and many errors. Therefore, the preferred title is "barrier attenuation mechanism", because it is more explicit about the hierarchy of barriers rather than all of them being included in the main path[36].” at line 216-227.
3. Based on your suggestion, the authors carefully review the work done throughout the whole paper and believe that this paper provided a safety evaluation system for the tailings dam stability. On the one hand, this paper developed a complete evaluation system from selecting evaluation indexes, constructing evaluation index system, applying TOPSIS model to evaluate subjects, and then using bow tie model to analyze countermeasures. On the other hand, the contents of the above parts are also relatively systematic and complete. In addition, the tailings dam stability evaluation software based on this evaluation system is also in development.

4. The calculation of seismic coefficient is illustrated in line 274-278, which is as follows: **The seismic coefficient is a safety factor derived from the comprehensive consideration of gravity, water content, infiltration line, pore water pressure, and slippage of the sliding surface when the dam experiences a magnitude 7 earthquake. The data, measured and calculated by engineers in the uranium tailings pond, could characterize the seismic capability of the dam.**

5. The value of "safety factor" is calculated by field engineers using Swedish circle method according to the evaluation index. The evaluation indexes referenced by the "safety factor" are basically the same as those shown in Table 1. The results show that the 9[#] dam is less than the 1[#] dam. The main reason is that the mechanical performance (shear strength) of the 1[#] dam is much higher than 9[#] dam and it can be seen from Table 1 that the mechanical properties of 9[#] dam perform very poorly in 10 dams. Mechanical performance is a very important evaluation index for dam stability, so the corresponding weight is also high. In addition, dam slope is also an important evaluation factor for the stability. It can be seen from table 1 that the dam slope of 9[#] dam is obviously steeper than that of 1[#] dam, which is very unfavorable to the stability. As is shown above, considering the evaluation indexes and weights, it can be concluded that the safety factor of 9[#] dam is less than that of 1[#] dam, which also verifies the evaluation results of this paper.

6. The concept of the contribution risk rate is based on the theory of safety system engineering to analyze the importance of each dam to the whole uranium tailings from the individuals and the whole. Specifically, it refers to the ratio of the instability risk rate of each dam to the whole uranium tailings pond risk (the sum of all dam body risk rates). we have added the instruction at line 328-329.

7. We appreciate your suggestion so much. It is true that comparison with more recent research results will make the effect better, but due to the particularity of uranium tailings ponds in multiple dam sections, there are relatively few relevant studies. Therefore, the evaluation results of this paper are compared with the latest dam monitoring data of this uranium tailings pond and briefly described, hoping to achieve the same effect. The specific modification is as follows: **The uranium tailings reservoir evaluated in this paper is in the treatment stage and the condition of the dam will change accordingly. According to the latest monitoring data of the uranium tailings dam, the evaluation results are still consistent with the results of this paper.**

For Reviewer 2:

We greatly appreciate your suggestions for this paper, especially for the introduction, Your suggestions make the paper more complete.

1. We added the rest of paper at the end of the introduction at line 103-106 as follow: **The accuracy of the method is verified by comparing the evaluation results with those obtained by the traditional Swedish circle method. And then after briefly analyzing the weak links using ABC analysis, the bow tie model is used to analyze the causes and propose safety countermeasures.**

2. We added some literature to enrich the introduction at line 81-99 as follow: **A. Hegde and Tanmoy Das combined pseudo-static seismic loading with the strength**

reduction analysis to check the seismic stability of the dam, and conducted nonlinear dynamic stability analysis to simulate the true earthquake events[16]. Florian Tatu and Dan Stematiu evaluate the safety of the tailings dam in Novat, and the unexpected rise of water table in the pond was the main cause of the dam failure. The results show that in both hypotheses, the hydraulic gradient at dam crest are higher than the critical hydraulic one (0.54), which leads to the internal erosion of the dam body failure[17]. In order to find a rapid evaluation method for the stability of rock slope, JIANG An-min et al. constructed a TOPSIS evaluation model based on entropy weight and made some improvements to meet the evaluation needs. Finally, four sections of slope are selected as examples to study, and the results are compared with those of improved grey relational evaluation and extension evaluation, and the three evaluation results are consistent[18]. DONG Fa et al. used strength reduction method to analyze the difference of slope stability under different rainfall and rainfall duration. The results show that there was a positive correlation between rainfall and the overall displacement of the slope, and the stability evaluation of the slope under rainfall conditions should take factors such as slope deformation characteristics, seepage law and safety factor into comprehensive consideration[19].

3. TOPSIS and VIKOR are typical MCDM. TOPSIS evaluation methods have much improved forms and are widely applied. The VIKOR method is rarely used and rarely has an extended form. VIKOR method requires the determination of criterion weight coefficient and criterion value, which is difficult to achieve in actual decision-making. Due to the fuzziness and uncertainty of a large number of decision-making problems, the criterion value and weighted value coefficient and other parameters cannot be completely determined. In addition, it is difficult for decision-maker to compare the importance of criteria in pairs, so the criteria weight coefficient cannot be determined by AHP, ANP, CNP and other methods. Finally, when VIKOR method is used for evaluation, it is necessary to combine more complicated fuzzy mathematical algorithms. This also limits the application of this method.

PROMETHEE is a multi-objective decision-making method based on a pair-wise comparison of schemes. It is a ranking method based on "outranking" relation. However, decision-makers are often inaccurate in grasping orderly relationships and priority strengths. In addition, this method is inferior to TOPSIS and AHP in structural analysis of problems. Similarly, the ELECTRE is based on outranking relation to eliminate the inferior scheme. Thus, the plan set is gradually narrowed down until the decision maker can choose the most satisfactory one. Except that ELECTRE has the same disadvantages as PROMETHEE, this method makes insufficient use of the decision matrix.

For Reviewer 3:

Thank you very much for your valuable suggestions, which also made us re-examine the whole paper. This paper is devoted to providing a complete, concise and effective evaluation system for the stability of uranium tailings dam with multiple dam sections. Uranium tailings are a small branch of the mining field. Due to its

special nature of containing a large amount of radionuclides, it has great significance for safety and environmental protection, but there are few related researches. In this paper, the stability indicators or safety evaluation points of many geotechnical dams, and other types of tailings dams are introduced into the stability evaluation of uranium tailings dams, hoping to provide new feasible suggestions for the decommissioning treatment of uranium tailings reservoirs. In addition, this paper emphasizes the factors that are easy to be ignored in the stability work of uranium tailings dam, and adds the following explanations: “Due to the cumulative effect of external forces, the dam is prone to generate loose structures with thixotropic and liquefaction. When the precipitation and pore water pressure are too large, the mechanical properties will be reduced and the dam will be unstable. It is easy to ignore this in practical engineering applications.” at line 271-275. Paying attention to the seepage parameters of the dam is very important to improve the stability of the uranium tailings dam.

Simple and effective evaluation methods and reliable index parameters have always been the core of evaluation, so the paper focuses on these two aspects. The evaluation method in this paper can not only have accurate and reasonable results, but also effectively distinguish the results that are difficult to distinguish, which is the advantage of combining variable weight with TOPSIS. In addition, another contribution of this paper is that the equalization function can be adjusted according to different evaluation objects in order to obtain the most appropriate and the most authentic evaluation results. This paper is hoped to arouse the interest of scholars in this field and apply the equalization function to a wider range of decision-making problems related to uranium tailings.

For Associate Editor:

Dear Editor,

We have responded in detail to the three reviewers' comments, especially on the topic of how to promote the stability of uranium tailings pond dam and the development of decommissioning treatment of uranium tailings pond in this paper and the responses are given in a different color (blue).

Finally, we would like to express our gratitude to the referees for their comments on the paper, which has improved our work and made this paper more perfect.

Besides, the corresponding author of this paper is JIANG Fuliang, the first author, and the corresponding author's Email is jfljfd@163.com. For the purpose of contributing to this paper, an author has been added to this paper. I hope the above two points could be modified and I apologize for any inconvenience caused.

Appendix B

Dear Editors and Reviewers:

Thanks very much for taking your time to review this manuscript. I really appreciate all reviewers' comments and suggestions! All of suggestions have enabled us to improve our work. Based on the instructions provided in your letter, we uploaded the file of the revised manuscript and our responses are given in a different color (red). The following is the reply to the referees' comments.

For Reviewer 4:

Thank you for your efforts. I have replied to your suggestion.

Modified “The main purposes of this study were to solve the problem of safety evaluation of tailings dam stability under the multiple factors” to “**The main purposes of this study is to analyse the evaluation of tailings dam stability under the multiple factors**” at line 11-12.

For Reviewer 5:

We greatly appreciate your suggestions for this paper, especially for the introduction, Your suggestions make the paper more complete.

1. Replace “in” with “of”
2. Modified “and propose safety measures to make the evaluation system more complete.” to “**and to propose safety measures to make the evaluation system more complete.**” at line 31-33;
3. added “to” after “belongs” at line 43 and used small letter at line 44;
4. Checked the spelling at line 82;
5. Modified “indexes” to “**indices**” at line 117;
6. Modified “Considering” to “**considering**” at line 146;
7. Modified “The evaluation matrix is determined by the two elements of the evaluation object to and the evaluation index.” to “**The evaluation matrix is determined by the two elements including the evaluation object and the evaluation index.**” at line 159;
8. Modified “the stability of the tailings ponds is related to the nearby residents’ safety and the Xiangjiang rivers’ sustainable development.” to “**the stability of the tailings ponds is consequent upon the nearby residents’ safety and the Xiangjiang rivers’ sustainability.**” at line 237-238;
9. Modified “The seepage capacity of the dam is represented by "seepage coefficient"” to “**"seepage coefficient" represents the seepage capacity of the dam**” at line 270;
10. Modified “the safety factors of 1[#] dam, 3[#] dam, 5[#] dam, 6[#] dam, 7[#] dam and 9[#] dam are very small” to “**the differences in safety factor values for 1[#] dam, 3[#] dam, 5[#] dam, 6[#] dam, 7[#] dam and 9[#] dam are very small**” at line 317-318;
11. In response to the recommendations made by line 237, the situation regarding the uranium tailings reservoir is classified;
12. Modified “the contribution risk rate of each dam” to “**the rate of risk contributed by each dam**” at line 317-318;
13. In response to the recommendations made by line 352, please see references [39]

and [40]

14. Modified “evaluation conclusion” to “**result obtained**” at line 396.

For Associate Editor:

Dear Editor,

We have responded in detail to the reviewers’ comments, and the certificate of language editing is shown below